# Deep Eutectic Solvents as Catalysts in the Synthesis of Active Pharmaceutical Ingredients and Precursors

Chiara Falcini and Gonzalo de Gonzalo *

Departamento de Química Orgánica, Universidad de Sevilla, c/Profesor García González 1, 41012 Sevilla, Spain; cfalcini@us.es
* Correspondence: gdegonzalo@us.es

**Abstract:** Deep Eutectic Solvents (DESs) have appeared in recent years as an appealing alternative to classical organic solvents, due to their valuable environmental properties. In addition, these compounds, formed by the combination of one hydrogen bond donor with a hydrogen bond acceptor at a defined stoichiometric ratio, present other valuable activities not only as a reaction medium. DESs can also be employed as catalysts through hydrogen-bond interactions in different chemical transformations, thus substituting hazardous reagents and solvents. The search for novel and more environmentally friendly catalysts is an area of interest of pharmaceutical chemists, and therefore, the efforts made in the application of DESs as catalysts in the synthesis of APIs or its precursors are described, focusing mainly on condensations, nucleophilic additions to carbonyl moieties, and multicomponent reactions.

**Keywords:** deep eutectic solvents; APIs; catalysis; multicomponent reactions; condensations

## 1. Introduction

The chemical industry, especially the pharmaceutical one, has a great impact on the environment. Several tons of waste are generated to produce Active Pharmaceutical Ingredients (APIs) and their precursors [1], thus leading to high E-factor values for these processes (from 25 to higher than 100), expressed as kg of waste generated by kg of product obtained [2,3]. The main part of these wastes is formed by the solvent(s), employed both as the reaction medium and for the product downstream. When referring to the synthesis of APIs and their precursors, it can be estimated that more than 80% of the wastes generated correspond to solvents [4]. As stated by the Twelve Principles of Green Chemistry [5,6], which establish a set of rules for the development of more sustainable chemical synthesis, the use of safer solvents is an area of main interest to develop sustainable procedures. Thus, classical organic solvents commonly employed in organic synthesis are not environmentally friendly, as they have relatively low boiling points and are highly flammable, toxic, and harmful for human health. The replacement of these hazardous solvents represents a challenge for chemists [7], with some more sustainable alternatives appearing in recent years such as biobased solvents [8–10], supercritical fluids [11,12], or neoteric solvents. In this last category, we can include ionic liquids (ILs) [13–15], widely used at the beginning of this century as solvents for several valuable transformations. Nowadays, studies performed on these compounds have shown that they present serious environmental issues, and Deep Eutectic Solvents (DESs) have appeared as a valuable alternative [16–20].

The term "Deep Eutectic Solvent" was introduced for the first time in 2003 by Abbott and collaborators [21], with the idea of preparing liquid mixtures at room temperature, using solid compounds with high melting points as the starting material. Since this seminal work, interest in this type of solvent has increased considerably both at the industrial and academic level, as demonstrated by the number of publications on this topic in the last

two decades. When they appeared in the literature, DESs were considered as a new class of ILs, due to their similar characteristics and properties. However, classic ILs and DESs represent two different types of solvents [22]. A DES, in fact, is a combination of two or more solid components which, through the formation of hydrogen bonds, create an eutectic mixture, defined as a mixture of substances that has a lower melting point than that of the individual components; by contrast, classic ILs are compounds completely composed of ions. In most cases, DESs are obtained by mixing a hydrogen bond acceptor (HBA), typically ammonium, phosphonium, and natural amino acid salts, with a hydrogen bond donor (HBD) such as sugars, polyols, ureas, and natural carboxylic acids; therefore, all raw materials can be derived from renewable sources [23–25].

DESs present valuable properties such as high thermal stability, low vapor pressure, very low toxicity, and high biodegradability. These solvents are easy to prepare, with a complete atom economy, presenting tunability of their physicochemical properties. Therefore, DESs are appropriate for employment in pharmaceutical chemistry with different purposes, thus also demonstrating a high versatility. Of course, DESs can be employed as solvents with improved properties for both the synthesis of APIs and precursors [26,27] and for the solubilization of drugs [28]. In addition, the recent development of the so-called therapeutic DESs (THEDES) has been observed, in which one of the components is a bioactive compound or a pharmaceutical ingredient which can be released in the appropriate environment [29]. Apart from these roles, DESs have demonstrated their activity as catalysts in different reactions [30–34]. As these compounds are formed by hydrogen bond donors and acceptors (Figure 1), they can mediate a wide set of transformations that occur through hydrogen bond catalysis, an area of high interest in organic chemistry [35–37]. Hydrogen bonding to an electrophile increases the reactivity of this compound, as it is more activated to a nucleophilic attack. Hydrogen bonding catalysis is widely found in nature and can be employed for electrophile activation in several reactions. Thus, a set of processes including cycloadditions, condensations, multicomponent reactions, and redox can be carried out in the presence of different amounts of DESs acting as hydrogen-bond catalysts and as cosolvents. In the present review, we have focused on the efforts made for the application of different DESs as catalysts in the synthesis of molecules with pharmaceutical interest.

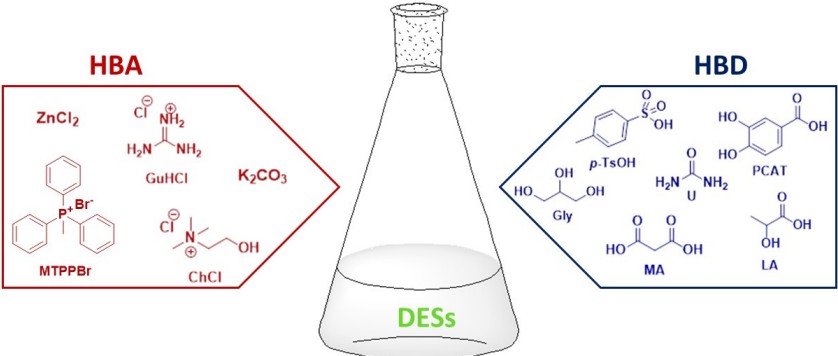

**Figure 1.** Structure of the hydrogen bond donor (HBD) and acceptor (HBA) components of the DESs employed as catalysts in the preparation of APIs and precursors.

## 2. DESs as Catalysts in Condensation Processes

β-Hydroxy-functionalized derivatives are valuable precursors to a wide variety of compounds such as nitroalkenes, 2-aminoalcohols, or 2-nitroketones. Among those β-hydroxy compounds, nitroaldols or β-nitroalcohols can be employed as building blocks in pharmaceutical chemistry [38] and as synthons for the synthesis of natural products [39]. Thus, β-nitroalcohols can be easily reduced to 1,2-amino alcohols which are contained in many APIs [40]. Generally, nitroaldols are prepared using the Henry reaction (nitroaldol reaction), involving C–C bond formation between a nucleophilic nitroalkane and an elec-

trophilic aldehyde or ketone, a process developed at room temperature in the presence of a base [41]. However, the use of bases often leads to the formation of side products. In 2012, Singh et al. employed DESs as catalysts in the Henry reaction for the first time [42]. The choline chloride/urea [ChCl:U] (1:2) DES was tested as a catalyst (20% *v/v*) in methanol as the reaction medium, due to the ability of urea of catalyzing this type of process. A set of nitroaldols containing different aromatic and heteroaromatic groups were synthesized with yields between 69 and 95% after 5–60 min (Scheme 1). The recycling of the DES in the reaction of 4-nitrobenzaldehyde with nitromethane was studied, and after five cycles, the yield was observed to only decrease from 95% to 90%. The authors also investigated the synthesis of other β-hydroxy derivatives including β-hydroxy nitriles and β-hydroxy carboxylic acids employing DESs as catalysts. Different percentages of DESs for both syntheses were analyzed, and it was observed that for the preparation of β-hydroxy nitriles, a 30% *v/v* DES led to a 95% yield after 2 h and, for the preparation of β-hydroxy carboxylic acids, the best DES percentage was 40% *v/v*, achieving a 91% yield after 3 h.

**Scheme 1.** Henry reaction between (hetero)aromatic aldehydes and nitromethane catalyzed by urea–based DESs.

α-Diazocarbonyl compounds are widely used as building blocks for organic synthesis [43]; moreover, they can be used for the structure-selective alkylation of DNA and RNA [44]. Among the different methodologies reported in the literature for the preparation of these compounds, the classic methodology consists of the condensation between aldehydes and acyldiazomethanes in the presence of a strong base such as LDA or sodium hydride. These methods require harsh conditions, and other synthetic routes employing heterogeneous catalysts have been investigated, but the work-up process in these reactions has presented some drawbacks. Thus, in 2017, Miraki et al. tested some DESs as catalysts and solvents in the aldol-type condensation of aromatic aldehydes containing electron-withdrawing or -donating groups, heterocyclic aldehydes, and aliphatic aldehydes with ethyl diazoacetate (EDA) at room temperature (Scheme 2) [45]. The syntheses were performed using potassium carbonate as the base mixed with glycerol to form the DES at different molar ratios, observing the best results in the presence of a 1:5 ratio of base to glycerol, leading to final product yields between 60 and 96%. When aromatic and heterocyclic aldehydes were tested, the reactions were completed after 2 h, whereas aliphatic aldehydes required 4 h. The use of an α,β-unsaturated aldehyde such as cinnamaldehyde led to the final product with moderate yield (73%).

**Scheme 2.** DES-catalyzed condensation of aldehydes with ethyl diazoacetate.

Methylthiotriazolo[1,5-a]pyrimidines are compounds with antimicrobial activity [46], and different synthetic approaches have been performed for their preparation. In most cases, long reaction times are required and difficult work-ups must be performed using non-environmentally friendly catalysts. Therefore, a novel approach has recently been developed by employing DESs as catalysts for the preparation of these compounds through

a condensation process between 4-dimethylamino-1-aryl-3-alken-2-ones and 3-amino-5-methylthio-1*H*-1,2,4-triazole [47], as shown in Scheme 3. DESs containing ChCl as a hydrogen bond acceptor and different hydrogen bond donors were tested in this reaction. It was observed that by using the DES formed by ChCl and *p*-toluensulfonic acid (*p*-TsOH) in a molar ratio of 1:2, the desired product can be obtained with 90% yield after 5 min at 120 °C. It is important to note that the molar ratio of the DES was essential for a high reactivity, as when ChCl:*p*-TsOH in a 1:1 ratio was tested, an incomplete conversion was observed. The reaction scope was checked by employing several enaminones as starting materials. After 5 min at 120 °C, excellent yields of the final cyclic products were achieved (82–98%). The authors also performed mechanistic studies on this reaction, establishing that acid catalysis and solvation effects on the activated complex determined the regioselectivity of the process.

**Scheme 3.** Synthesis of methylthiotriazolo[1,5-a]pyrimidines by a DES-catalyzed condensation between 4-dimethylamino-1-aryl-3-alken-2-ones and 3-amino-5-methylthio-1*H*-1,2,4-triazole.

## 3. DESs as Catalysts in Multicomponent Reactions

Multicomponent reactions (MCRs) are powerful synthetic methodologies that involve the simultaneous reaction of three or more reactants to produce a single, more complex product. MCRs offer several advantages in organic synthesis, such as high atom economy, step economy, and a rapid generation of molecular complexity. These reactions have become popular in medicinal chemistry, combinatorial chemistry, and the synthesis of complex natural products [48–50]. Since their discovery at the beginning of this century, DESs have been described as mediators for this type of reaction.

2,3-Dihydroquinazolinones are heterocyclic compounds widely used in the preparation of pharmaceuticals such as metolazone, quinethazone, raltitrexed, and fenquizone, due to the biological and pharmaceutical activities that these heterocycles present [51,52]. Generally, these molecules are synthesized by a multicomponent reaction between isatoic anhydride, an aldehyde or a ketone, and aromatic amines, using acid catalysts in organic media. This procedure presents several limitations, due to the use of non-environmentally friendly catalysts and solvents, strong acidic conditions, prolonged reaction times, and low yields. In 2012, Lobo et al. tested different DESs at 20% *v/v* as catalysts in the reaction of 4-chlorobenzaldehyde with 4-methylaniline and isaotic anhydride, developing a more sustainable process [53]. Among the eutectic solvents analyzed were ChCl:malonic acid (MA), ChCl:glycerol (Gly), ChCl:U, and ChCl:ZnCl$_2$, using methanol as reaction media at 65 °C for 2 h (Scheme 4A). The best result was achieved in the presence of ChCl:MA (1:1) with a 94% yield. The use of this DES was extended for the preparation of other 2,3-dihydroquinazolin-4(1*H*)-ones starting from aromatic and heteroaromatic aldehydes and different anilines, achieving yields from 80 to 95% at short reaction times (2–3 h). DES could be recycled up to four times through methanol evaporation after each reaction, with no loss in system activity.

**A**

R$_1$;R$_2$: Aryl, heteroaryl

80–95% yield

**B**

R$_1$: Alkyl, aryl, heteroaryl

92–98% yield

**C**

91% yield

**Scheme 4.** (**A**) Synthesis of 2,3-dihydroquinazolinones through an MCR between isatoic anhydride, aldehydes, and aromatic amines catalyzed by ChCl:MA (1:1). (**B**) Preparation of 2,3-dihydroquinazolinones using ammonium acetate as nitrogen source and U:ZnCl$_2$ (3.5:1) as catalyst. (**C**) Synthesis of tryptanthrin catalyzed by U:ZnCl$_2$ (3.5:1) using isatoic anhydride and isatin.

The use of DESs as catalysts and solvents for the synthesis of 2,3-dihydroquinazolinones was further investigated in 2019 by Peña-Solórzano et al. [54]. A set of DESs were tested in the reaction between an aldehyde (benzaldehyde) and isatoic anhydride in the presence of a nitrogen source (NH$_4$OAc) at different temperatures (60–110 °C) for 5–120 min. The best condition was at 60 °C for 5 min in the presence of urea/ZnCl$_2$ (3.5:1) as the DES, thus obtaining a 97% yield in the desired product (Scheme 4B). The authors established that the better performance of the DES can be attributed to the Lewis characteristic of the mixture that enhances the electrophilicity of both carbonyl groups present in the reaction medium (isatoic anhydride and aldehyde). The reaction was extended to the synthesis of other 2,3-dihydroquinazolin-4(1*H*)-ones bearing different substituents, achieving excellent yields after short reaction times in all cases.

The authors finally described the preparation of the biologically active dihydroquinazolinones. Following the previously described methodology using aliphatic aldehydes, ammonium acetate and isatoic anhydride did not afford the final compounds, due to the formation of several side products. Thus, for the preparation of the alkaloid tryptanthrin, the reaction between isatoic anhydride and isatin for 30 min catalyzed by the DES led to the final product with high yield (91%, Scheme 4C) [54].

In 2013, He and coworkers described the use of a Brönsted-type DES (choline chloride:*p*-TsOH acid 1:1) for the one-pot synthesis of 2*H*-indazolo[2.1.*b*]phthalazinetriones [55], which are compounds employed as anticonvulsants, cardiotonics, and vasorelaxants [56]. Among the different synthetic methodologies for their preparation, the MCR of phthalhydrazide, an aldehyde and dimedone in the presence of a catalyst have recently gained strong interest. *p*-TsOH can be employed as a catalyst for this process, but it is deliquescent and difficult to recover. Therefore, the DES formed by ChCl/*p*-TsOH (1:1) was prepared and employed as the catalyst for this reaction. The best result was achieved when the MCR was developed in methanol containing 15% *v/v* of the DES at reflux for 4 h, leading

to an 88% yield of the benzofused product. The synthetic methodology in the optimized conditions was extended to the preparation of a wide set of final products starting from different substituted aromatic aldehydes, with yields from 85 to 93% (Scheme 5).

X: H, F, Cl, Br, OCH$_3$, NO$_2$

85–93% yield

**Scheme 5.** MCR for the preparation of 2*H*-indazolo[2.1.b]phthalazinetriones starting from phthalhydrazide, aldehydes, and dimedone in the presence of ChCl:*p*-TsOH (1:1) as catalyst.

The Groebke–Blackburn–Bienayme (GBB) reaction consists of an MCR between an aldehyde, a 2-aminoazide, and an isocyanide catalyzed by an acid, yielding the final fused imidazole derivatives [57]. One type of such derivatives are the imidazo[1,2-a]pyridines, which are valuable structures present in several compounds with pharmaceutical activity including zolpidem, alpidem, and zolimidine [58]. Initial studies were carried out in the GBB reaction of benzaldehyde, 2-aminopyridine, and cyclohexyl isocyanide in the presence of DESs presenting different compositions, achieving the final product with the highest yield when using ChCl:Urea (1:2) at 90 °C, as the final product was recovered with 80% yield [59] (Scheme 6). These best conditions were applied for the GBB reaction employing different aldehydes, 2-aminoheterocycles and iscocyanides. The reactions proceeded with good yields after short times (1–2 h) when using 2-aminopyrazines and substituted 2-aminopyridines. A wide set of aromatic aldehydes presenting electron-donating or electron-withdrawing groups can also be employed in this procedure, and even heteroaromatic and polycyclic aldehydes are suitable for the reaction, leading to high yields. Alkyl isocyanides can also be employed while maintaining good activities. DES recycling was studied in the model reaction. After one process was completed, the reaction mixture was dissolved in water and the product was extracted with ethyl acetate (EtOAc). Evaporation of the water at 80 °C allowed the DES to be recovered for a further cycle. After four cycles, only a slight decrease in the reaction yield was observed.

X: CH, N
Y: H, Br
R$_1$: H, CH$_3$
R$_2$: Cyclohexyl, $^t$Bu
R$_3$: Aryl, heteroaryl

80–94% yield

**Scheme 6.** Groebke–Blackburn–Bienayme reaction for synthesizing imidazo[1,2-a]pyridines using ChCl:U (1:2) as catalyst.

3,4-Dihydropyrimidin-2(1*H*)-ones or thiones (DHPMs) are valuable molecules that can be employed as antibacterials, antivirals, calcium antagonists, and antihypertensives [60]. These compounds can be prepared through a Biginelli reaction [61], employing β-ketoesters, aldehydes, and (thio)urea in the presence of different catalysts, but in almost all the cases, sensitive conditions and hazardous reagents are required [62]. In 2019, a novel catalytic

approach was developed by using DESs for mediating this transformation [63], as shown in Scheme 7. Several ChCl-based DESs, presenting different types of hydrogen bond donors including *p*-TsOH, trichloroacetic acid (TCA), monochloroacetic acid (MCA), propionic acid (PA), and ethylene glycol (EG), were tested in the reaction of benzaldehyde with ethyl acetoacetate and urea at 70 °C. After 40 min, an 88% yield of the final product was achieved in the presence of ChCl:*p*-TsOH (1:2). The amount of catalysts was analyzed, and a ratio of 0.6 mmol of DES per 2.0 mmol of aldehyde and ethyl acetoacetate was observed, which was optimal in order to obtain the highest yields. If the DES amount increased, the yield decreased, probably due to the high acidity of the catalyst. The scope of the reaction was extended to other aldehydes and β-ketoesters, observing that it was suitable for a wide set of aromatic aldehydes, different β-diketonates, and urea or thiourea to obtain DHPMs with generally high yields (78–92%), except for ethyl benzoylacetate, for which no reactivity was observed. The reaction was also tested with aliphatic aldehydes, observing a 52% yield in the process between *n*-butylaldehyde, ethyl acetoacetate, and ureas, whereas a 48% yield was achieved in the presence of hexaldehyde. The recycling of ChCl:*p*-TsOH (1:2) was analyzed in the model reaction with benzaldehyde, ethyl acetate, and urea at 70 °C, evaporating the solvent after each stage and employing the reused DES for a further cycle.

X: H, F, Cl, Br, $OCH_3$, $NO_2$
Y: O, S
$R_1$: $CH_3$, Ph
$R_2$: OMe, OEt

85–93% yield

**Scheme 7.** Biginelli reaction for the preparation of DHPMs catalyzed by ChCl:*p*-TsOH (1:2).

The pyrazolo[3,4-b]pyridine moiety appears in a wide set of valuable compounds [64,65]. A class of these compounds, the spirooxindoles, are part of natural alkaloids and scaffolds for the development of antivirals [66]. In 2019, their synthesis was described by using DESs as catalysts in an MCR involving 1*H*-pyrazol-5-amine, an isatin, and an enolizable C/H-activated compound in the presence of microwave irradiation [67]. Different choline-chloride-based DESs were tested in the model reaction, consisting of the treatment of 1,3-diphenyl-1*H*-pyrazol-5-amine, isatin, and 5,5-dimethylcyclohexane-1,3-dione at equimolar quantities under 500 W of irradiation at 60 °C (Scheme 8). The highest yields were observed in the presence of a lactic acid (LA)-based DES, focusing on ChCl:LA. When the ChCl:LA ratio was 1:1, an 86% yield was observed after 20 min, whereas the use of ChCl:LA (1:2) afforded the desired product in 95% yield at the same time. This reaction was scaled up to 100 mmol, maintaining such excellent results. The authors also performed recycling of the catalysts, observing only a slight decrease in the reaction yield (78%) after eight cycles. A set of isatins reacted with 1,3-diphenyl-1*H*-pyrazol-5-amine and 5,5-dimethylcyclohexane-1,3-dione, leading to the final compounds with high yields, despite the presence of different substituents in the starting materials. Also, 1*H*-pyrazol-5-amines were tolerated in this reaction. Other enolizable C−H-activated compounds such as barbituric acid, cyclopentane-1,3-dione, cyclohexane-1,3-dione, and 1,3-indanedione were analyzed in this reaction, recovering the corresponding pyrazolo[3,4-b]quinoline spirooxindoles in excellent yields.

**Scheme 8.** Three-component reaction between 1*H*-pyrazol-5-amines, isatins, and enolizable C/H molecules to achieve spirooxindoles using ChCl:LA (1:2) as catalyst.

Benzofused seven-membered heterocycles such as 1,4-benzodiazepines and 1,4-benzoxazepines present strong interest in medicinal chemistry due to their biological properties [68,69]. Some examples of these structures are diazepam and chlordiazepoxide (anti-anxiety drugs), and other compounds present activity such as histamine receptor agonists, calcium agonists, and analgesics. In 2016, a study described the preparation of these compounds via a three-component reaction involving an *o*-phenylenediamine or 2-aminophenol, 2-dimedone, and some aromatic aldehydes in the presence of ChCl:U (1:2) as a catalyst and solvent [70] (Scheme 9). In general, benzodiazepines were achieved in higher yields (80–94%) and shorter reaction times than the benzoxazepines (68–88% yields). It was also observed that the use of aromatic aldehydes containing electron-withdrawing groups led to better results with respect to those with electron-donating groups. The DES can be simply recycled by dissolving the reaction mixture in water and filtering the insoluble product. After water evaporation, the DES was employed again, showing that it maintains its activity for four cycles.

**Scheme 9.** Three-component reaction for the preparation of benzodiazepines and benzoxazepines catalyzed by ChCl:U (1:2).

Imidazoindoles and pyrroloimidazoles are structures that present a wide set of pharmacological properties, being employed as antimicrobial, antiviral, or immunosuppressive compounds [71,72]. These molecules can be prepared using Pd-catalyzed reactions, but this method presents some limitations. Some DESs have been recently employed as catalysts and solvents in the three-component MCR procedure using an aldehyde, a 1,3-diketone, and hydantoin, achieving the best result when using guanidinium chloride (GuHCl)/urea (1:2) at 80 °C [73], as shown in Scheme 10. After 30 min, a 95% yield of the desired compound was obtained. A recycling study of the DES showed that the solvent/catalyst can be employed for six reactions without loss in catalytic activity.

**Scheme 10.** Model reaction for the three-component preparation of a pyrroloimidazole between hydantoin, an aldehyde, and a 1,3-diketone catalyzed by GuHCl:U (1:2).

In 2015, Hu et al. described the DES-catalyzed synthesis of a set of functionalized pyrroles through a multicomponent reaction involving four starting materials [74]. The pyrrole moiety is present in several relevant compounds, including APIs [75]. Therefore, several approaches have been performed through their synthesis, with MCRs also being described. In 2010, a paper described the preparation of highly functionalized pyrroles by combining 1,3-dicarbonyl compounds, amines, aromatic aldehydes, and nitroalkanes in the presence of $FeCl_3$ as a catalyst [76]. This method presents some drawbacks such as low yields and tedious work-up procedures, with other catalytic methodologies being developed with better yields using transition metals [77] or other non-metallic catalysts [78]. All the processes described required long reaction times. Therefore, the MCR of aniline, 4-chlorobenzaldehyde, acetylacetone, and nitromethane was tested in the presence of ChCl:MA (1:1) as a catalyst and solvent (Scheme 11). At 80 °C, 88% of the final pyrrole was recovered after short reaction times. The reaction was scaled up to 50 mmol, observing an 89% yield after 1 h. The scope of the reaction was extended to other amines, aldehydes, and 1,3-dicarbonyl compounds. A wide range of amines can be employed, leading to moderate-to-good yields. Aromatic and heteroaromatic aldehydes also afforded the final compounds with good results, but unfortunately, alkyl aldehydes led to complex reaction mixtures. Also, a set of 1,3-dicarbonyl compounds including methyl, ethyl or 2-methoxyethyl acetoacetate, isobutyl 3-oxobutanoate, and methyl 3-oxopentanoate butyl acetate were analyzed, achieved high yields. DES recycling for the model reaction was studied, observing that the reaction can be conducted for five cycles with only a minimal loss in the yield (84% vs. 88%).

**Scheme 11.** Preparation of functionalized pyrroles in a four-component reaction catalyzed by ChCl:MA (1:1).

The pyranopyrimidine moiety is present in a wide set of compounds presenting pharmacological activity (antibronchitis, antitumoral, antihypertensive, or antimicrobial) [79,80]. Several methodologies have been described for their synthesis [81,82], but in 2023, the synthesis of compounds presenting a pyranopyrimidine ring out in a multicomponent reaction catalyzed by DESs was described (Scheme 12). Thus, a DES formed by methyltriphenyl-phosphonium bromide (MTPPBr) and 3,4-dihydroxybenzoic acid (protocatechuic acid, PCAT) at molar ratio 1:1 was employed in the reaction between barbituric acid, 4-hydroxyhydrocoumarin, and

benzaldehyde to yield the corresponding chromenopyranopyrimidine [83]. Using 1.5 mmol of DES at 80 °C, a 95% yield of the final product was achieved after short reaction times. The reaction was extended to other aromatic aldehydes, obtaining yields from 85 to 95% in all cases. DES recycling was studied, adding ethanol and water after each reaction completion, and filtering the reaction to separate the DES. Only a slight decrease was observed in the system, recovering the final product with 83% yield after the fourth cycle.

X: Cl, OH, COOH, CH$_3$, NO$_2$                    85–95% yield

**Scheme 12.** Preparation of functionalized pyranopyrimidines in a three-component reaction involving barbituric acid, 4-hydroxyhydrocoumarin, and aromatic aldehydes catalyzed by DES.

During the catalytic process, barbituric acid is converted into its enol by the DES and reacts with the activated aldehyde with the loss of one water molecule to form an intermediate that reacts as a Michael acceptor with 4-hydroxycoumarin to form a second intermediate, which suffers from intramolecular cyclization and the loss of water to yield the desired compound.

## 4. Other Reactions Involving Carbon-Heteroatom Bond Formation

Thioethers present interest in biological and pharmaceutical chemistry ], as they can be easily converted into sulfoxides, which is a moiety present in a wide set of APIs [84]. Generally, thioethers are obtained by aromatic nucleophilic substitution reactions (S$_N$Ar) [85], through the coupling between thiolates and aryl halides, the reduction of aryl sulfones or sulfoxides, or by transition-metal-catalyzed reactions [86]. All these procedures are not sustainable, due to the involvement of toxic reagents and solvents, high temperatures, and long reaction times. In 2017, Pant and Shankarling performed the synthesis of thioethers in DESs by choosing the S$_N$Ar reaction between 1-chloro-4-nitrobenzene and *p*-thiocresol as the model process [87] (Scheme 13). Different choline-chloride-based DESs were tested as catalysts and/or solvents. The yields obtained were below 50% when employing oxalic acid, malonic acid, tartaric acid, or glycerol as an HBD. By contrast, the use of a urea-based DES afforded a 98% yield. To prove the role of ChCl:U (1:2) as a catalyst, a fructose/urea DES as well as ChCl or urea alone were tested in the reaction, observing unsatisfactory results. Different percentages of ChCl:U (1:2) were analyzed, observing that 20% *v/v* is the most effective catalyst loading. The best conversions were achieved at 80 °C in the presence of polar solvents such as acetonitrile and ethanol. A set of aryl chlorides with electron-withdrawing groups were tested in the reactions as well as different alkyl thiols, obtaining yields between 88 and 98% after short reaction times. Finally, the recyclability of the DES was studied when the reaction between 1-chloro-4-nitrobenzene and *p*-thiocresol was scaled up to 20 mmol. After four cycles, the yield was still 94%.

Due to their valuable properties, amines can be employed in compounds with pharmaceutical activity, including antihypertensive, antihistamine, and anti-inflammatory drugs, with their synthesis being a target of interest for organic chemists [88]. In 2011, the selective N-alkylation of aromatic primary amines by different types of catalysts was described, including lipases and DESs [89]. Thus, the N-alkylation of aniline with hexyl bromide led to the alkylated product in 78% yield after 4 h using ChCl:U (1:2) at 50 °C, as shown in Scheme 14. A set of aromatic amines was tested, achieving in all cases good yields in reaction times ranging from 2 to 10 h in the presence of hexyl, butyl, or benzyl bromide.

The authors propose that the hydrogen bond interactions between the DES and the aromatic amine increase the nucleophilicity of these amines, thus making its reaction faster with the alkyl bromide. The recycling of ChCl:U (1:2) showed that it can be employed five times with only a slight decrease in the catalytic activity.

R₁: NO₂, CHO, COOH, CN
R₂: H, CH₃

**Scheme 13.** ChCl:U-catalyzed synthesis of thioethers starting from aryl chlorides and aromatic thiols.

R: Aryl

**Scheme 14.** N-alkylation of anilines with hexyl bromide catalyzed by ChCl:U (1:2) for the preparation of secondary amines.

In 2019, the preparation of methaqualone, mecloqualone, and analogs in good yields was described starting from benzoxazine and aniline [54]. In order to synthesize the corresponding amines, U:ZnCl₂ was employed as a catalyst at a ratio of 3.5:1 (Scheme 15).

X: Cl, CH₃.

**Scheme 15.** Synthesis of methaqualone (X: *o*-CH₃) and analogs in presence of urea:ZnCl₂ (3.5:1) as catalyst.

Amides play a key role in the synthesis of biologically active molecules [90], usually being produced by the reaction between amines and acyl chlorides, acid anhydrides, esters, or carbodiimides [91]. These compounds can also be obtained by the metal-mediated hydrolysis of organonitriles [92], but all these procedures generate a lot of toxic wastes. In 2014, Patil et al. described the preparation of a set of primary amides starting from aldehydes or nitriles using choline-chloride-based DESs as a catalyst [93]. They studied the reaction between benzaldehyde and hydroxyl amine hydrochloride in the presence of a ChCl:ZnCl₂ (1:2) as a catalyst and solvent at 100 °C for 15 h, obtaining a 94% yield. The procedure was extended to other aromatic and aliphatic aldehydes, leading in all cases to yields between 86 and 94%.

The synthesis of primary amides starting from nitriles and water as substrates was performed at 100 °C for 12 h using ChCl:ZnCl₂ (1:2) as a catalyst, achieving a 98% yield (Scheme 16). Aromatic nitriles bearing electron-donating or -withdrawing groups afforded yields between 93 and 98%, and when an aliphatic nitrile was used, this value decreased to 90%. DES recycling was also analyzed, observing that after five cycles, the yield decreased from 94% to 70% using aldehydes, whereas in the case of employing nitriles, the yield was 82% after five cycles.

**Scheme 16.** Synthesis of aromatic amides starting from nitriles catalyzed by a zinc chloride-based DES.

Carbamate-bearing compounds can be employed in the synthesis of drugs such as retigabine or albendazole [94]. Generally, carbamates are synthesized from amines and alcohols requiring the use of toxic phosgene or its derivatives [95]. To avoid the use of these compounds, some methods have been developed employing reagents such as isocyanates, carbonates, azides, CO, or $CO_2$ [96]. In general, these procedures require long reaction times, high temperatures and pressures, and high molar ratios of the carbonylation agents. In 2018, a novel route to prepare carbamates employing DESs as a catalyst and solvent was developed by Inaloo and Majnooni [97]. The reaction between N,N-*o*-diphenylurea and 1-propanol was studied in ChCl:ZnCl$_2$ (1:2) as a DES with different molar ratios of reagents, temperatures, and reaction times (Scheme 17). The best results were obtained with a molar ratio of urea/alcohol of 2:1, at 120 °C within 18 h, obtaining 83% of the final carbamate. Afterward, other DESs were tested, but in all cases, lower yields were achieved regarding ChCl:ZnCl$_2$ (1:2). This procedure was extended to several substrates such as ureas presenting aromatic and aliphatic substituents as well as different types of primary, secondary, and tertiary alcohols and phenols. Regarding the alcohols used, the best results were obtained when the primary ones were employed, with yields around 80–85%. Those ureas bearing electron-withdrawing groups led to the highest yields in the carbamate synthesis (86–89%). DES recycling showed that its catalytic activity was maintained for five cycles, with only a 3% decrease in the yields.

**Scheme 17.** DES-catalyzed preparation of carbamates starting from N,N-*o*-diphenylurea and alcohols.

## 5. Conclusions

Since they were described for the first time in 2003, Deep Eutectic Solvents have represented a versatile class of compounds, with several applications in different chemical fields. Apart from being excellent and sustainable solvents for organic synthesis procedures, DESs have also demonstrated their ability for catalyzing different reactions through hydrogen bond interactions, mainly focused on carbon–carbon bond formations or multicomponent processes. DESs, with their unique properties, have emerged as versatile and effective catalysts across a spectrum of chemical transformations.

While the potential of DESs as catalysts is evident, challenges such as viscosity, toxicity, and scalability must be addressed to facilitate widespread industrial adoption. Future research should focus on overcoming these hurdles and expanding the scope of DES catalysis.

We can conclude that the catalytic potential of DESs for the preparation of APIs and their precursors represents a paradigm shift in the way we approach chemical transformations. From their origin as solvents to their evolution into catalysts, DESs have demonstrated the ability to reshape traditional synthetic pathways toward greener and more sustainable methodologies. As research in this area continues to unfold, the catalytic landscape is likely to witness further innovations and applications, marking DESs as catalysts at the forefront of sustainable chemical synthesis.

**Author Contributions:** Conceptualization, G.d.G.; methodology, C.F. and G.d.G.; writing—original draft preparation, C.F. and G.d.G.; writing—review and editing, C.F. and G.d.G.; funding acquisition, G.d.G. All authors have read and agreed to the published version of the manuscript.

**Funding:** This project has received funding from the EU's Horizon Europe Doctoral Network Program under the Marie Skłodowska-Curie grant agreement no. 101072731.

**Data Availability Statement:** No new data were created.

**Conflicts of Interest:** The authors declare no conflicts of interest.

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
