# Peer review of "Deep Eutectic Solvents as Catalysts in the Synthesis of Active Pharmaceutical Ingredients and Precursors"

_catalysts, doi:10.3390/catal14020120_

Round 1

Reviewer 1 Report

Comments and Suggestions for Authors

The referee has ever reviewed this paper and was satisfied with the revision. Thus, I support its publication at the current stage.

Comments on the Quality of English Language

none

Author Response

We want to thank the referee for his/her kind comments.

Reviewer 2 Report

Comments and Suggestions for Authors

The review focuses on the consideration of catalytic reactions involving DESs the synthesis of active pharmaceutical ingredients and precursors. In general, the manuscript is written in an easy to understand language and should undoubtedly be of interest to readers in the relative fields. The following comments are available.

1. The authors present the point of view according to which ionic liquids and DES are considered different classes of substances. There is (and has also been actively supplemented recently) an alternative idea of a different classification of such substances and, which is that the classic ILs (mentioned by the authors) can be attributed to the first generation (https://doi.org/10.1002/smsc.202200048).And the next generation includes DESs; poly(ionic liquids), PILs (for example, http://dx.doi.org/10.1039/d3gc02131a) ionic liquid crystals, ILCs; (https://doi.org/10.1016/j.nanoen.2022.108087); redox-active ionic liquids, RAILs. A completely new family is also worth mentioning here: polymeric deep eutectic solvents, PDES (https://doi.org/10.1016/j.molliq.2023.122677). But, on the other hand, any classification is conditional and incomplete, and whether it is worth mentioning these recent works, it is possible to leave this decision entirely to the consideration of the authors.

2. The main sections of the manuscript (2-4) are mainly provided with links to works up to 2018. Perhaps the authors should pay attention to this.

3. There are relatively recent reviews close to the subject of the manuscript (for example, https://doi.org/10.3390/molecules26206286; https://doi.org/10.1016/j.indcrop.2022.114990; https://doi.org/10.1038/s41598-023-45352-4). It would be interesting to know the authors' opinion on the differences between the manuscript and these works, as well as on the possible mention of its in the body of the manuscript.

4. Figure 1 has no reference in the text of the manuscript itself.

5. The paragraph (Lines 186-193) goes without the reference, although it is logically clear that it is a continuation of the previous paragraph, but nevertheless it is recommended to duplicate the reference here again.

Author Response

The review focuses on the consideration of catalytic reactions involving DESs the synthesis of active pharmaceutical ingredients and precursors. In general, the manuscript is written in an easy to understand language and should undoubtedly be of interest to readers in the relative fields. The following comments are available.

  1. The authors present the point of view according to which ionic liquids and DES are considered different classes of substances. There is (and has also been actively supplemented recently) an alternative idea of a different classification of such substances and, which is that the classic ILs (mentioned by the authors) can be attributed to the first generation (https://doi.org/10.1002/smsc.202200048).And the next generation includes DESs; poly(ionic liquids), PILs (for example, http://dx.doi.org/10.1039/d3gc02131a) ionic liquid crystals, ILCs; (https://doi.org/10.1016/j.nanoen.2022.108087); redox-active ionic liquids, RAILs. A completely new family is also worth mentioning here: polymeric deep eutectic solvents, PDES (https://doi.org/10.1016/j.molliq.2023.122677). But, on the other hand, any classification is conditional and incomplete, and whether it is worth mentioning these recent works, it is possible to leave this decision entirely to the consideration of the authors.

Referee is right in this statement. We would like to point the difference between the classic ionic liquids, formed mainly by ions, and the DESs, composed by mixing hydrogen bond donors and aceeptors. It is true that the concept IL has evolved to cover different types of neoteric solvents, as the referee indicated. By this reason, we have included the term “classic” to indicate the difference between those ILs and the DESs. At this point, it is not the scope of the revision to go further in the IL classification, as we are only focussing in DESs as catalysts.

  1. The main sections of the manuscript (2-4) are mainly provided with links to works up to 2018. Perhaps the authors should pay attention to this.

Referee is right, and also answering in part what is expressed in comment 3, we are focusing in the application of all type DESs as catalysts for the preparation of APIs and their precursors. It is true that there are references older than 2018, but they are valuable to show in which type of reactions DES-catalysts have been employed. In order to clarify this a bit more, we have deleted the term “recent” at line 75.

  1. There are relatively recent reviews close to the subject of the manuscript (for example, https://doi.org/10.3390/molecules26206286; https://doi.org/10.1016/j.indcrop.2022.114990; https://doi.org/10.1038/s41598-023-45352-4). It would be interesting to know the authors' opinion on the differences between the manuscript and these works, as well as on the possible mention of its in the body of the manuscript.

Referee is right mentioning that some revisisions on the use of DESs as catalysts. In this one, we have focused in this relatively novel area by mentioning sepecifically the application of DES as catalysts in the synthesis of pharmacologically active molecules. We have included this specific search in the abstract and in the introduction, as well as in the conclusions. Referee has indicated us some research manuscripts focused in the application of DESs as catalysts. The first one, a review focused on the use of choline chloride-based DES in pharmaceutical processes (not only DES as catalysts, but also as solvents and with other purposes) is our reference 32.  As we have indicated that the scope of the revision is the use of these catalysts in the preparation of compounds with pharmaceutical acitivity, we have included the last one, focused on the preparation of pyranopyrimidines catalyzed by DESs.

  1. Figure 1 has no reference in the text of the manuscript itself.

Referee is right, we have included Figure 1 in the text.

  1. The paragraph (Lines 186-193) goes without the reference, although it is logically clear that it is a continuation of the previous paragraph, but nevertheless it is recommended to duplicate the reference here again.

We have rewritten that paragraph following the indications of referee 3, so we have separated it in two paragraphs. Now, as stated by the referee, we have included the references when necessary.

Reviewer 3 Report

Comments and Suggestions for Authors

Authors presented valuable data on green catalytic medium and important reactions in the presence of Deep Eutectic Solvents. A variety of building blocks to pharmaceutically active compounds can be obtained by applying such a medium. Data on components of DES, their ratio and reaction conditions are included in the review and discussed, benefits of DESs are presented quite well. The prospects of synthetic procedures in the medium of DES are highlighted.

The manuscript can be published after minor revision. Some comments:

1   1) Mistakes in the Schemes 3 and 8. Nitrogen atom is missed in the structure of triazolopyrimidine (Scheme 3). In the Scheme 8 please check: ketoester R1C(O)CH2C(O)OR2 as reagent and substituent COOR2 in the position 5 of pyrimidine ring OR beta-diketone as reagent and C(O)R group in pyrimidine product.

2   2) Title to the Chapter 2 is not so perfect, because the process demonstrated in the Scheme 3 does not include C-C bond formation, during this condensation two C-N bond formation takes place.

3   3)  Scheme 5 describes two-component process, from formal point of view both reactions ate not multi-component. I would suggest providing Scheme 4 with several processes based on isatoic anhydride: a) with R1C(O)H and R2NH2, b) with R1C(O)H and NH4OH, c) with Ar-C(O)H and AcONH4; d) with isatine. The process shown in the Scheme 5A could be moved in the Chapter 4.

4 4)  Line 182 – what is 1, may be 2,3-dihydroquinazlin?

5  5) It would be better to outline in the Abstract: in what types of transformations DESs represent a valuable alternative to traditional synthetic methods (nucleophilic addition to carbonyl group, cyclocondensations, some multicomponent processes.

6   6)  Line 448 -please delete “software, X”.

Author Response

We thank referee 3 for the valoration made on the manuscript. We have proceed in the following way regarding the comments performed:

1   1) Mistakes in the Schemes 3 and 8. Nitrogen atom is missed in the structure of triazolopyrimidine (Scheme 3). In the Scheme 8 please check: ketoester R1C(O)CH2C(O)OR2 as reagent and substituent COOR2 in the position 5 of pyrimidine ring OR beta-diketone as reagent and C(O)R group in pyrimidine product.

Schemes 3 and 8 have been corrected as indicated by the referees. We have used ketoester as reagent in the example, so COOR2 was indicated in the pyrimidine ring.

2   2) Title to the Chapter 2 is not so perfect, because the process demonstrated in the Scheme 3 does not include C-C bond formation, during this condensation two C-N bond formation takes place.

Referee is right, by this reason we have entitles section 2 as “DESs as catalysts in condensation processes”

3   3)  Scheme 5 describes two-component process, from formal point of view both reactions ate not multi-component. I would suggest providing Scheme 4 with several processes based on isatoic anhydride: a) with R1C(O)H and R2NH2, b) with R1C(O)H and NH4OH, c) with Ar-C(O)H and AcONH4; d) with isatine. The process shown in the Scheme 5A could be moved in the Chapter 4.

We have followed the referre’s suggestion splitting Scheme 5 into 2. Schem 5B was incorporated to Scheme 4 together with Scheme 4.B, that is the preparation of the final compounds employing ammonium acetate. There was a mistake in the text, as NH4OH was not employed, so this has been also corrected. Scheme 5.A was moved to section 4 as Scheme 15, as is a process for the formation of an amine.

4 4)  Line 182 – what is 1, may be 2,3-dihydroquinazlin?

Referee is right, it should state 2,3-dihidroquinazolin-4(1H)-ones

5  5) It would be better to outline in the Abstract: in what types of transformations DESs represent a valuable alternative to traditional synthetic methods (nucleophilic addition to carbonyl group, cyclocondensations, some multicomponent processes.

As referee stated, we have incorporated a sentence in the abstract indicating which processes are mainly catalysed by the DES for the preparation of APIs and precursors.

6   6)  Line 448 -please delete “software, X”.

“Software, X” has been removed.

Round 2

Reviewer 2 Report

Comments and Suggestions for Authors

The manuscript can be accepted in the current version. Unfortunately, it is a bit confusing due to a bit of quoting of recent works (2018-2023) in the main parts of the paper for the literature review, which will be published in 2024.